# A Generative Approach to Person Reidentification

**DOI:** 10.3390/s24041240

**Published:** 2024-02-15

**Authors:** Andrea Asperti, Salvatore Fiorilla, Lorenzo Orsini

**Affiliations:** Department of Informatics—Science and Engineering (DISI), University of Bologna, 40126 Bologna, Italy; salvatore.fiorilla@unibo.it (S.F.); lorenzo.orsini4@studio.unibo.it (L.O.)

**Keywords:** person re-identification, image generation, diffusion models, latent space, representation learning

## Abstract

Person Re-identification is the task of recognizing comparable subjects across a network of nonoverlapping cameras. This is typically achieved by extracting from the source image a vector of characteristic features of the specific person captured by the camera. Learning a good set of robust, invariant and discriminative features is a complex task, often leveraging contrastive learning. In this article, we explore a different approach, learning the representation of an individual as the conditioning information required to generate images of the specific person starting from random noise. In this way we decouple the identity of the individual from any other information relative to a specific instance (pose, background, etc.), allowing interesting transformations from one identity to another. As generative models, we use the recent diffusion models that have already proven their sensibility to conditioning in many different contexts. The results presented in this article serve as a proof-of-concept. While our current performance on common benchmarks is lower than state-of-the-art techniques, the approach is intriguing and rich of innovative insights, suggesting a wide range of potential improvements along various lines of investigation.

## 1. Introduction

The challenge of person re-identification lies in recognizing comparable subjects across a network of non-overlapping cameras, which is a common scenario in multi-camera surveillance systems [1]. In its typical formulation, the person re-ID task aims to identify a specific individual from an extensive collection of person images, known as the *gallery*, by utilizing a *query* image [2]. While this task falls under the framework of image retrieval problems, its unique objective of determining the identity of the person within a query image, typically expressed by a distinctive *ID* label, introduces intriguing peculiarities.

At its core, person re-ID involves acquiring knowledge about distinct characteristics of individuals, enabling the differentiation between images depicting the same person and those featuring different individuals. The difficulty of this task stems from substantial variations in viewpoint, pose, lighting, and image quality across diverse cameras in real-world scenarios. In such situations, individuals may appear in multiple cameras across various locations, thereby intensifying the challenge of feature acquisition [3].

The typical approach to the re-ID task involves a two-phase process: a representation learning phase, where interesting and distinctive features of individuals are extracted, and a metric learning phase, where training objectives are designed using various loss functions or sampling strategies. This process aims to shape the embedding space in a way suitable for searching and retrieval.

Both phases are complex and very interesting. For feature extraction, modern approaches have leveraged deep learning techniques, exploiting convolutional networks [4,5], Generative Adversarial Networks [6,7,8,9], Visual Transformers [10,11,12,13,14], and different kinds of attention mechanisms [15,16,17,18]. Features can be enhanced by exploiting part-based methods, focusing on extracting features from specific regions of interest [10,19,20], pretraining [21,22], and multitasking [23] via a suitable set of pretext tasks.

Metric Learning is typically based on some form of contrastive learning [24], either using contrastive loss [25,26,27,28] or, more frequently, triplet loss [29,30,31,32]. Both approaches aim to encourage similar examples to be close in the feature space and dissimilar examples to be separated. In the context of triplet loss, ref. [33] introduced a focal loss to the triplet criterion, assigning more weight to negative samples compared to positives. Additionally, computing centroids, such as the mean of all samples belonging to each instance, helps with cluster creations as well [34,35,36,37]. The triplets can be extended to a quadruplet loss, with two negative samples and one positive pair, to enlarge the inter-class variations [38] or to N-tuple loss for joint optimization of multiple instances [39]. An extensive amount of works use a form of contrastive loss in combination with the softmax loss [40,41], i.e., recent works are [19,23,42]. Contrastive learning can also be used in combination with techniques such as generative learning [43], multi-instance learning [44], and spatial attention [45].

Contrary to all such methods, we adopted a distinct approach by associating a unique latent representation with each identity. This representation was learned as the conditioning information necessary to generate diversified samples of the particular person from random Gaussian noise (see Figure 1a). The generator takes as input a (learned) embedding of the identity emb(id), a noise vector *z*, and produces a sample image GEN(emb(id),z). The next step involves computing a right inverse $GEN^{-1}$ to the generator, producing, from an individual’s image *X*, an embedded identity $id_{X}$ and a noise vector $z_{X}$, such that $GEN(id_{X},z_{X})=X$ (see Figure 1b). The idea is that $id_{X}$ should capture the interesting and individual features of the person represented in *X*, while the “noise” $z_{X}^{\prime }$ should contain all information unrelated to the specific identity, such as background, pose, etc. The identity $id_{X}$ is then used to retrieve samples from the gallery by comparing their embeddings in the latent space.

The inversion of the generator was the most delicate part of the work and is discussed in Section 4.

As the generative model, we employed an instance of the recent diffusion models [46,47] due to their remarkable sensitivity to conditioning, unbiased sampling capabilities, and excellent expressiveness. Examples of the generation of images conditioned on the individual identity are given in Figure 2.

The generative quality is not perfect, and can probably be improved upon. In particular, for this prototyping work, we used a low-resolution version of the dataset, with images of a 64×32 dimension (instead of the usual 128×64). The embedding space for the identities has 32 dimensions. In spite of being so small, the conditioning seems to work well, as exemplified in Figure 2.

The results presented in this article represent a preliminary investigation in the direction described above. They are primarily intended as a proof of concept, testing the feasibility of the approach. The current performance, measured on standard benchmarks such as Market-1501, is encouraging, though it falls short of various state-of-the-art approaches.

### 1.1. Achievements

The goal of this work was to answer two conceptual questions, namely:Is it possible to characterize a person’s identity (from the person re-ID perspective) as the conditional information required to generate a diversified set of images relative to the given person?In that case, is it possible to compute a right inverse to the generator, extracting from an image of a person their conditioning identity?

Although, from the point of view of person re-identification, our results are slightly below the state of the art, our work provides evidence that both questions are likely to have a positive answer.

### 1.2. Structure of the Work

The article is structured as follows. Section 2 discusses related work. Section 3 introduces the specific class of generative models utilized in this study, namely Denoising Diffusion Models, with a focus on a particular subclass known as Implicit Models. The methodology, including the inversion of the generative network, is detailed in Section 4. Section 5 provides a comprehensive description of the neural network architecture. A brief summary of our results is presented in Section 6. Section 7 explores the latent space of identities, examining the spatial organization of latent representations and the significance of different variation factors. Section 8 is dedicated to ablation studies and discusses alternative solutions that were considered but ultimately not adopted. Concluding remarks and directions for future research are provided in Section 9.

## 2. Related Work

In the introduction, we provided an overview of the primary methodologies used in addressing the person re-identification task. Given that this paper adopts a generative paradigm, this section focuses on exploring relevant literature within this specific context.

Generative techniques in person re-identification have primarily been utilized for data augmentation and unsupervised domain adaptation (UDA). The majority of these works employ Generative Adversarial Networks (GANs), which have only recently been challenged by the emergence of diffusion models.

Domain adaptation methods utilize style transformation techniques to adapt images from a source domain to resemble those of a target domain. This transformation allows the images to retain the content of the source domain while adopting the visual style of the target domain. The new images obtained in this way are then used to fine-tune the network parameters. GAN-based image style transfer has quickly become a popular method for transferring knowledge between source and target domains in unsupervised cross-domain person re-identification, as evidenced by [6,7,48,49].

CycleGANs, as discussed in [9,50], are employed for this purpose. In the first mentioned work, a target encoder is used to obtain a discriminative mapping of target images into the transformed source feature space, thereby fooling a domain discriminator. The role of the discriminator is to distinguish between the features of the target domain and those of the transformed source domain. In [50], the objective is to swap features of the source and target environments to generate cross-domain images that preserve identity-related features associated with the source (or target) background features. This process is subsequently reversed to reconstruct the original input image, facilitating a self-supervised cyclic generation.

Chen et al. [43] propose an alternative approach in their study. They employ a 3D mesh-guided GAN to generate views, which are then integrated with original images through memory-based contrastive learning, aiming to achieve view-invariant representations of instances. In this methodology, training occurs jointly; the GAN learns the data distribution through adversarial training, while the contrastive instance discriminator acquires representations by retrieving each instance from a pool of candidates.

In [8], GANs fulfill a dual role. They utilize CycleGAN and Siamese networks to transfer image styles between domains while ensuring pedestrian identities are preserved. This is followed by iterative self-training with GANs to enhance both global and local features within the target domain, thereby facilitating robust feature learning in unlabeled data.

In [51], the introduction of the PCDS-GAN model marks a significant advancement in synthesizing source-labeled images against domain-specific backgrounds, effectively bridging the domain gap and enhancing domain adaptation efficiency. This is accomplished by separating pedestrian images into foreground, background, and style features. A U-Net-based Hole-Filling Module (HFM) is employed for transferring the background between different domains, tasked with reconstructing the scene areas previously occupied by the source foreground. This technique enables the generation of person images set against a variety of target domain backgrounds.

## 3. Denoising Diffusion Models

In this section, we provide a concise overview of diffusion models, a topic extensively covered in existing literature. Readers familiar with this area may opt to skip this section.

Denoising Diffusion Models (DDMs) [46] mark a significant breakthrough in deep generative modeling, posing a challenge to the longstanding predominance of Generative Adversarial Networks [52]. These models have been applied with great success in various recent and notable projects, such as [53,54,55]. Key factors contributing to their rising prominence include unparalleled generation quality, high adaptability and ease of conditioning, the ability to generate diverse and robust samples, stable training processes, and remarkable scalability.

An important characteristic of this generative paradigm is its robust probabilistic foundation. However, a formal exploration of the underlying theory behind denoising diffusion models is beyond the scope of this article. For a deeper theoretical understanding, we refer the reader to the extensive literature available [46,47]. Here, we aim to provide an operationally focused description of the denoising model, comprehensible even without a deep understanding of its theoretical background.

In very rough terms, a diffusion model trains a single network to denoise images with a parametric amount of noise. This network is then used to generate new samples in an iterative manner, starting from a purely ’noisy’ image and progressively removing a decreasing amount of noise.

This process is traditionally referred to as *reverse diffusion*, as it aims to “invert” the *direct diffusion* process, which consists of iteratively adding noise (see Figure 3).

We concentrated on a particular subclass of diffusion models, known as Implicit Diffusion Models [47], which have been a focal point of our research in numerous previous works. A key feature of these models is their fully deterministic reverse diffusion process. This attribute is crucial for applications that require embedding the output back into its latent representation, as discussed in [57]. Another significant advantage of Implicit Diffusion Models is their efficiency, typically requiring only a minimal number of iterations (about 10, as reported in [58,59])—a stark contrast to other techniques that may need thousands of iterations. Additionally, their remarkable conditioning capabilities have been demonstrated in the field of precipitation forecasting [56], where the goal is to predict the probability distribution of a specific atmospheric parameter based on recent weather conditions.

### 3.1. The Denoising Network

The only trainable component of the reverse diffusion process is a *denoising network* $\epsilon_{\theta }\left(x_{t},\alpha_{t},c\right)$, which takes as input a noisy image $x_{t}$, a signal rate $\alpha_{t}$, possibly a condition *c*, and tries to guess the noise present in the image. The model is trained in a completely supervised way. The main steps are the following:pick a random image $x_{0}$ from the train set, coherent with the condition *c*;select a random step *t* of the process; to each step *t* is associated a signal rate $\alpha_{t}$ defined by a suitable noise scheduling (more in the sampling section);sample a random Gaussian noise ϵ;create a corrupted image as a weighted combination of $x_{0}$ and ϵ:
$$x_{t}=\sqrt{\alpha_{t}}x_{0}+\sqrt{1\;-\;\alpha_{t}}\epsilon$$train the network to properly guess the amount of noise present in $x_{t}$, by minimizing the distance between $\epsilon (x_{t},\alpha_{t},c)$ and ϵ.

The previous steps are summarized in the pseudocode of Algorithm 1, where $q(x_{0}|c)$ denotes the distribution of training data, relative to the condition *c*.
**Algorithm 1** Training1: **repeat**2:       $x_{0}∼q\left(x_{0}|c\right)$                ▹ take a sample coherent with *c*3:       t∼Uniform(1,…,T)                   ▹ choose a timestep4:       ϵ∼N(0;I)                ▹ create random Gaussian noise5:       $x_{t}=\sqrt{\alpha_{t}}x_{0}+\sqrt{1\;-\;\alpha_{t}}\epsilon$        ▹ corrupt the sample with signal rate $\alpha_{t}$6:       Take a gradient descent step on ${\left|\left|\epsilon -\epsilon_{\theta }\left(x_{t},\alpha_{t},c\right)\right|\right|}^{2}$  ▹ backpropagate the loss7: **until** converged

The approach to conditional training previously described was formally examined in [60]. The paper proposes a blend of conditional and unconditional generation, a method some researchers found beneficial for training, though we observed no significant improvements.

Another strategy for implementing conditioning involves using an auxiliary classifier [52], akin in concept to AC-GANs [61]. This method trains a classifier $f_{\phi }\left(c|x_{t}\right)$ on a noisy image $x_{t}$ to predict its class *c*. The gradient $\nabla_{x}\log f_{\phi }\left(c|x_{t}\right)$ is then utilized to steer the diffusion sampling process, enhancing the generation of samples that more accurately reflect the given condition. This technique remains untested in our work but represents a potential avenue for future enhancements.

The preferred architecture for the denoising network adopts a U-Net structure [62]. Our specific version is discussed in Section 5.

To heighten the network’s sensitivity to the signal rate $\alpha_{t}$, this parameter is often encoded through a bespoke sinusoidal transformation, dividing it into a series of frequencies, mirroring the technique of positional encodings in Transformers [63].

### 3.2. Sampling

Sampling is an iterative process. Starting from a purely noisy image $x_{T}$, we progressively remove noise by calling the denoising network. Specifically, if the error predicted by the network at timestep *t* is $\epsilon =\epsilon (x_{t},\alpha_{t},c)$, then the corresponding denoised prediction is
$$\hat{x_{0}}=\frac{1}{\sqrt{\alpha_{t}}}\left(x_{t}-\sqrt{1\;-\;\alpha_{t}}\epsilon \right)$$

Next, ϵ is re-injected into the network at signal rate $\alpha_{t-1}$, to obtain the next noisy image:$$x_{t-1}=\sqrt{\alpha_{t-1}}{\tilde{x}}_{0}+\sqrt{1-\alpha_{t-1}}\epsilon$$

The pseudocode of the sampling process is given in Algorithm 2.
**Algorithm 2** Sampling1: $x_{T}∼N\left(0,I\right)$2: **for** t=T,...,1 **do**3:      $\epsilon =\epsilon_{\theta }\left(x_{t},\alpha_{t},c\right)$                ▹ predict noise4:      ${\tilde{x}}_{0}=\frac{1}{\sqrt{\alpha_{t}}}\left(x_{t}-\sqrt{1-\alpha_{t}}\epsilon \right)$     ▹ compute denoised result5:      $x_{t-1}=\sqrt{\alpha_{t-1}}{\tilde{x}}_{0}+\sqrt{1-\alpha_{t-1}}\epsilon$  ▹ re-inject noise at rate $\alpha_{t-1}$6: **end for**

A significant component of the reverse diffusion process is the scheduling of diffusion noise ${\alpha_{t}}_{t=1}^{T}$. The authors in [46] suggested the use of linear or quadratic schedules. However, such choices lead to a rapid decrease in the initial steps, creating challenges in the generation phase. The literature offers alternative scheduling functions that decrease more gradually, including the “cosine” or “continuous cosine” schedule [64,65]. Opting for a gentler scheduler not only solves the generation issues but also reduces the number of iterations needed. For our purposes, we have implemented a generator with 10 diffusion steps.

## 4. Methodology

As introduced earlier, the core concept is to assign a unique latent representation to each identity, which is learned as the necessary conditioning information for generating specific individual samples from random Gaussian noise. The overall structure of the generative model is depicted in Figure 1a. This model is an example of a diffusion model, as discussed in the preceding section.

The subsequent step involves reversing the generative model, as shown in Figure 1b. This process enables us to derive, from an individual’s image, the latent representation of their identity and noise that captures all details unrelated to identity, such as background and pose. The latent identity holds significance for person re-identification because comparing it with the latent representations of gallery samples allows us to identify the most similar identities.

The feasibility of inverting diffusion models through appropriately trained neural networks has been explored in many of our prior works. The overall idea was presented in [57]; this approach was applied in [58] for the “reification” of artistic portraits by embedding a portrait into a latent space of human faces and reconstructing the closest real approximation. In [59], the diffusion inversion was utilized to create a trajectory in the latent space that induces a smooth rotation effect on human faces.

In this work, after several trials, we decided to divide the inversion networks into two separate models, as illustrated in Figure 4. One model, named *Img2ID*, converts images into latent identities, while the other, named *Img2Noise*, generates the noise. Given that both models extract orthogonal features from the source image, there was no compelling justification for them to share layers.

There are a few different possibilities for training the inversion models. Suppose we have a labelled image $\langle id,X_{id}\rangle$. We can train Img2ID to minimize the distance:(1)$$∥emb\left(id\right),img2ID\left(X_{id}\right)∥$$
where emb is the identity embedder of the generator. Similarly, Img2Noise can be trained to minimize the error:(2)$$∥GEN\left(Img2Noise\left(X_{id}\right),emb\left(id\right)\right),X_{id}∥$$
as the noise synthesized by $Img2Noise(X_{id})$ should result in the generation of $X_{id}$ when conditioned over emb(id). One issue with this approach is that the Market1501 dataset is relatively small. When constraining the inversion to images from the training set, where identities are disjoint from the gallery, the technique struggles to generalize effectively to new identities.

Given that we have a generator at our disposal, a natural idea is to leverage it to expand the training data. We select a random identity from the latent space of identities, along with a random noise ϵ, and generate an image $X_{id}=Gen\left(\epsilon ,id\right)$. We then proceed as previously described. However, a challenge arises from not knowing the distribution of identities in the latent space, which could be arbitrary, making it difficult to select a random identity. Our solution involves utilizing an auxiliary ID generator to learn the distribution of identities in the latent space, enabling us to sample effectively within this space.

Results of the inversion process are shown in Figure 5.

The reconstruction is satisfactory, although not flawless. Some details are lost, which could pose challenges for person re-ID.

For those interested, we also display the noise extracted from the image, which ideally contains all information not specific to the individual (middle row). The noise, assumed to have a Gaussian distribution, has been clipped within the range [−2.5, 2.5] and then renormalized to [0, 1] before visualization.

The inverted generator obtained through this process is extremely robust. In principle, it enables the derivation of the internal representation of any individual within any context, provided that the individuals and contexts are akin to those encountered in the training set.

## 5. Neural Network Architectures

In Section 3, it was clarified that the only trainable component within a diffusion model is the denoising network $\epsilon_{\theta }\left(x_{t},\alpha_{t},c\right)$. This network receives inputs including a noise rate $\alpha_{t}$, an image $x_{t}$ corrupted with a corresponding level of noise, and a condition *c*, which in our scenario represents the identifier of an instance. The objective is to predict the noise present in the image.

This task aligns with the conventional image-to-image paradigm, and the denoising network is typically constructed using the well-established U-net architecture [62]. The architecture of our implemented denoising network is illustrated in Figure 6; the main constituent blocks are depicted in Figure 7.

As is customary for U-nets, our network follows an encoder–decoder structure with skip connections that link corresponding layers in the encoder and decoder. This architectural choice facilitates the assimilation of both global and local structures within the image, making it highly compatible with the demands of the diffusion model.

The U-Net typically relies on parameters such as the number of downsampling blocks and the number of channels for each block, with the upsampling structure being symmetric. In our implementation, we opted for four downsampling blocks with respective channel dimensions [48, 96, 192, 384].

We emphasize that our work deals with images at a resolution of 64×32; for higher-resolution images such as 128×64, an additional layer in the U-chain may be necessary.

The input pertaining to the noise variance $\alpha_{t}$ is incorporated using sinusoidal positional embeddings. Subsequently, this information is vectorized, concatenated, and forwarded through the blocks along with the conditioning information of the ID and the processing of the noisy image.

Regarding the inversion networks, Img2Noise entails an image-to-image process, as the noise maintains the same spatial dimension as the input image. Based on prior studies [57], the U-Net architecture has demonstrated superior performance for such tasks.

The Img2Id model follows a similar approach, employing only the encoder portion of the U-Net. The processing is then finalized through a compact sequence of dense layers.

## 6. Evaluation

We conducted experiments utilizing the Market-1501 dataset [66], which comprises images of 1501 individuals captured by six cameras situated in front of an outdoor supermarket at Tsinghua University. The dataset includes 32,668 annotated bounding boxes. The training set comprises 12,936 images featuring 751 unique identities, while the gallery set contains 19,732 images encompassing 751 distinct identities. The query set consists of 3368 images, with 750 identities consistent with those in the gallery set.

The current performance of our technique falls somewhat below the state of the art. On Market1501, we achieved a mean average precision (mAP) of 73%, whereas state-of-the-art techniques attain a MAP of 96% in the supervised setting and approximately 90% in the unsupervised setting. The mAP score was computed following the guidelines provided by the dataset.

Visually, the results are satisfactory, and the errors made by the model are quite understandable given the low resolution we are working with. Distinguishing between certain identities at this resolution presents a significant challenge (see Figure 8).

## 7. Latent Space Exploration

In our approach, latent representations are synthesized as the features of each individual able to shape the generation of distinctive images of the person under different noises. As is typical in generative processes, this naturally leads to the creation of a highly informed and well-structured latent space, where similar persons yield similar encodings.

Figure 9 presents some examples; in the first column, we have several identities, and in the corresponding rows, we display five different individuals whose latent representations are closest, in terms of Euclidean distance, to the representation of the given identity. These images are sourced from the traditional Market1501 dataset [67], which we primarily used for our experiments. Notably, we observe the significant similarity between different identities, underscoring the non-trivial nature of the reidentification problem.

In Figure 10, we show the two closest identities, at a minimum distance of 0.14, and the two more apart from each other, at a maximum distance of 0.99. The average Euclidean distance is around 0.6.

Another intriguing operation involves deciphering how different explanatory factors of variation in the data are captured within the latent representation, akin to understanding the meaning of nucleotide sequences in a genome. However, these factors can often be entangled, meaning that a visible effect may depend on a combination of latent variables rather than a single one.

Our studies in this direction are very preliminary. As an illustration, in Figure 11, we demonstrate the effect on generation when modifying a single variable in the latent space of identities.

It is noteworthy to observe how the modification of a single variable can correspond to a significantly different identity.

Furthermore, investigating interesting semantic trajectories within the latent space can also facilitate the synthesis of new and distinct identities, which can be valuable, for instance, for data augmentation purposes.

## 8. Ablation and Alternatives

Several iterations of the aforementioned architecture have undergone testing.

Prior to delving into learning the distribution of latent variables using an auxiliary generator, we experimented with various regularization techniques to shape its distribution. This ranged from straightforward methods such as employing BatchNormalization layers to injecting a mild amount of noise. Subsequently, we explored a comprehensive variational model that should enforce the adoption of a Gaussian-shaped latent space through Kullback–Leibler regularization.

However, a drawback of the latter approach is the relatively weak signal influencing the definition of latent representations. This poses a challenge in balancing it with the KL component, as highlighted in [68]; in such a situation, the KL component could dominate, leading to the well-known variable collapse phenomenon [69].

The utilization of an auxiliary generator can essentially be conceptualized as a two-stage generative model [68,70]. Initially, we experimented with a variational autoencoder for implementing the auxiliary generator. However, this endeavor resulted in a notable increase in reconstruction loss, which corresponded to a loss of variance in generated data [71]. In contrast, a diffusion model appears to learn the distribution much more effectively in this regard.

Exploring a different research direction, we sought to increase the separation between distinct latent representations through the incorporation of a "repulsive" loss, which was proportionate to the inverse of their distance in the space, akin to a form of magnetic repulsion. This was complemented by a weak attractive loss, which encouraged points to remain close to the origin, effectively centering the latent space around it.

While we retained the latter, the repulsive loss did not appear to enhance performance and could potentially be detrimental. Our hypothesis is that the loss primarily impacts latent variables with minimal significance, attempting to segregate them across different subjects. This segregation is artificial and lacks semantic predictability, rendering it unreconstructable from the input images.

Finally, we conducted several experiments on the Embedding module for identities. In the current implementation, it comprises a single Keras Embedding layer. Initially, this layer expands labels to their categorical encoding and subsequently maps them to a latent space of the desired dimension through a single dense layer.

In principle, this setup should not impose any restrictions, as any embedding can potentially be learned from a categorical description. However, what remains unclear is the model’s flexibility to adapt and enhance encodings during training, along with the acquisition of additional knowledge. To explore this aspect, we experimented with more complex embedding modules, although they did not yield any discernible benefits.

## 9. Conclusions

The work presented in this article is first of all a *divertissment*; we aimed to test the conditional capabilities of generative diffusion models in a complex scenario. Specifically, we sought to learn the latent representation of different individuals as the shared information necessary to condition the generation of images of the given person from varying noise. This approach enables the separation of an individual’s identity from other instance-specific information, such as pose and background, expressed as part of the noise.

There are numerous avenues for improvement in this work. Both the generator and the network used for its inversion could be enhanced. A natural research direction involves repeating the experiments with images at the original 128 × 68 spatial resolution, potentially expanding the dimension of the latent space. Additionally, a detailed analysis of the errors made during the re-identification process could lead to valuable insights and potential enhancements. Leveraging the generator for intelligent and targeted data augmentation presents another intriguing possibility.

A limitation of our approach is its reliance on supervised training; to train the conditional generator, a significant number of instances of the same identity in different poses and contexts are required. Exploring whether the methodology can be extended to an unsupervised scenario presents a complex and interesting challenge. It is plausible that typical unsupervised person re-ID approaches, predominantly based on clustering, centroids, and pseudo-labels [72,73,74], could be adapted to our approach.

## Figures and Tables

**Figure 1 sensors-24-01240-f001:**
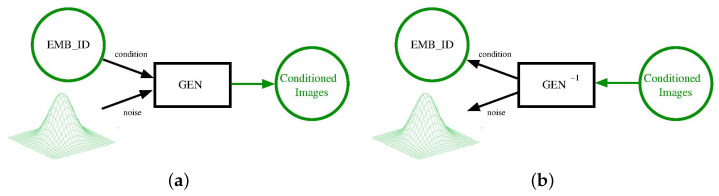
Conditional generative model (**a**) and its inversion (**b**). We trained a diffusion generative model conditioned on an embedding of people’s identity. The embedding of identity labels was learned along with the generator. In reverting the conditional generator, we obtained a model that was able to split the image into the identity of the person, and all of the contextual information (pose, background, and so on) that was mapped back to the noise.

**Figure 2 sensors-24-01240-f002:**
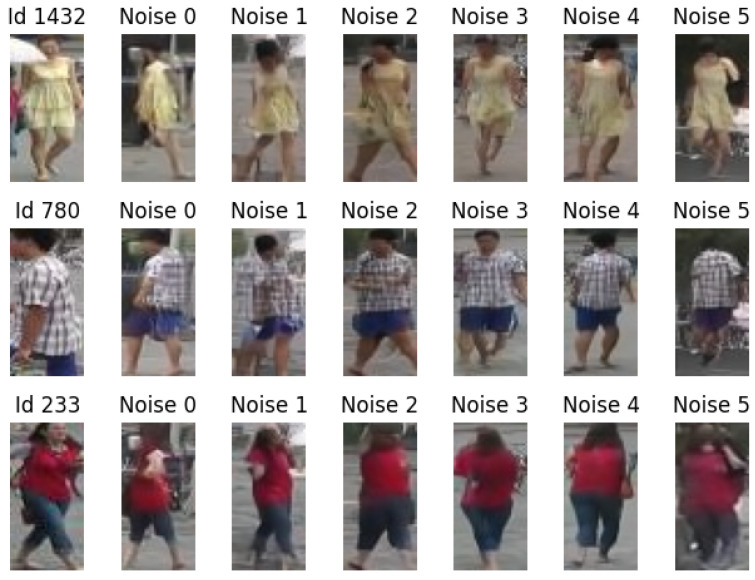
Conditional generation. On the left, three sample identities taken from the Market1501 dataset. We sample 5 random noises, and for each of them we generate an image conditioned by the given identity; the noise in each column is always the same.

**Figure 3 sensors-24-01240-f003:**
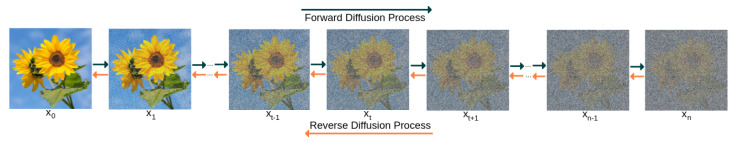
Direct and reverse diffusion. Picture from [56].

**Figure 4 sensors-24-01240-f004:**
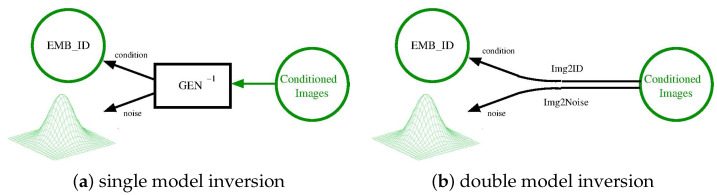
The inversion of the generator can be realized through a single network (**a**), or split in two distinct networks (**b**).

**Figure 5 sensors-24-01240-f005:**
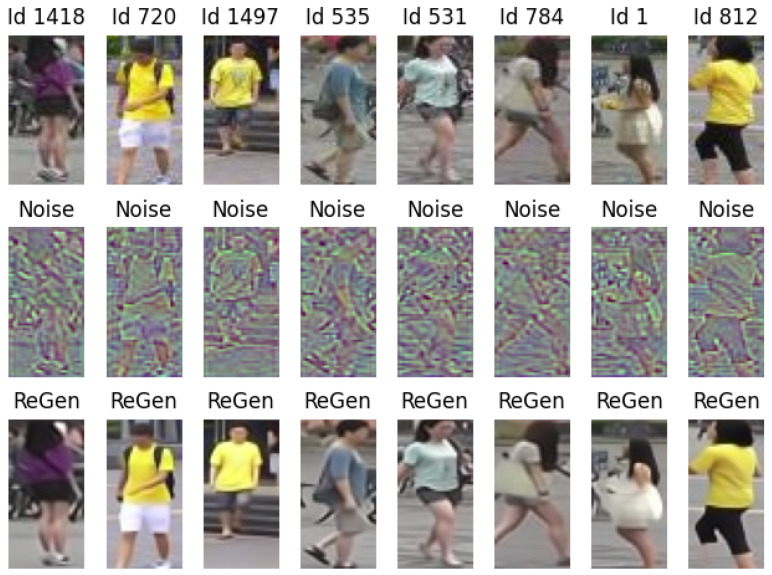
Reconstruction examples. In the fist row, we have original gallery images, in the middle row the noise synthesized by inversion, and in the third row the reconstructed images.

**Figure 6 sensors-24-01240-f006:**
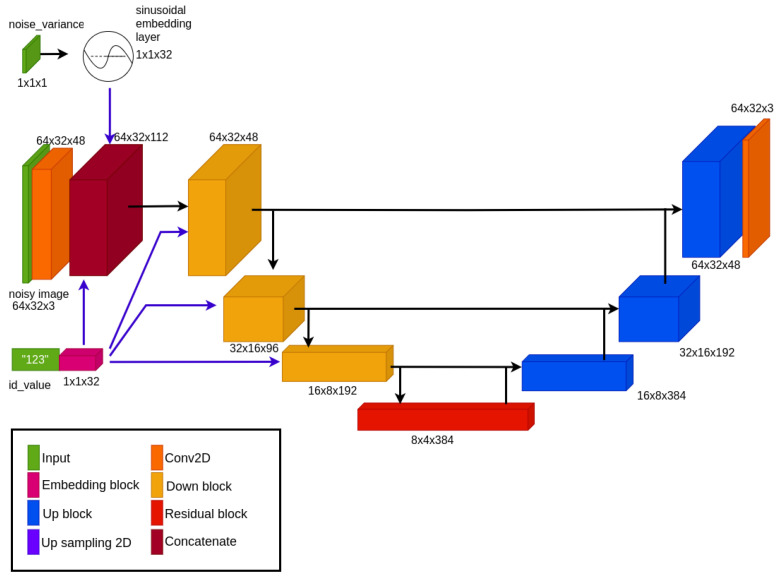
Denoising network architecture.

**Figure 7 sensors-24-01240-f007:**
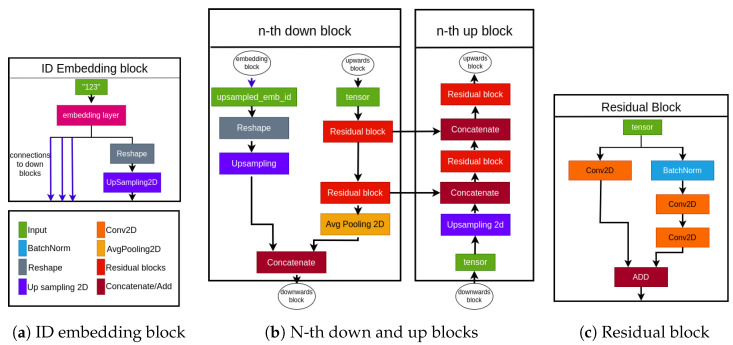
Main blocks of the Denoising Network. (**a**): The input ID undergoes processing through an embedding layer; the resulting output, after reshaping and upsampling, is concatenated with the various downblocks. (**b**): The nth downblock receives two inputs: the embedded ID and the output of the previous block. The block outputs two elements: the skipping connections for the upblock at the same level and the output for the next block. (**c**): a traditional Residual Block.

**Figure 8 sensors-24-01240-f008:**
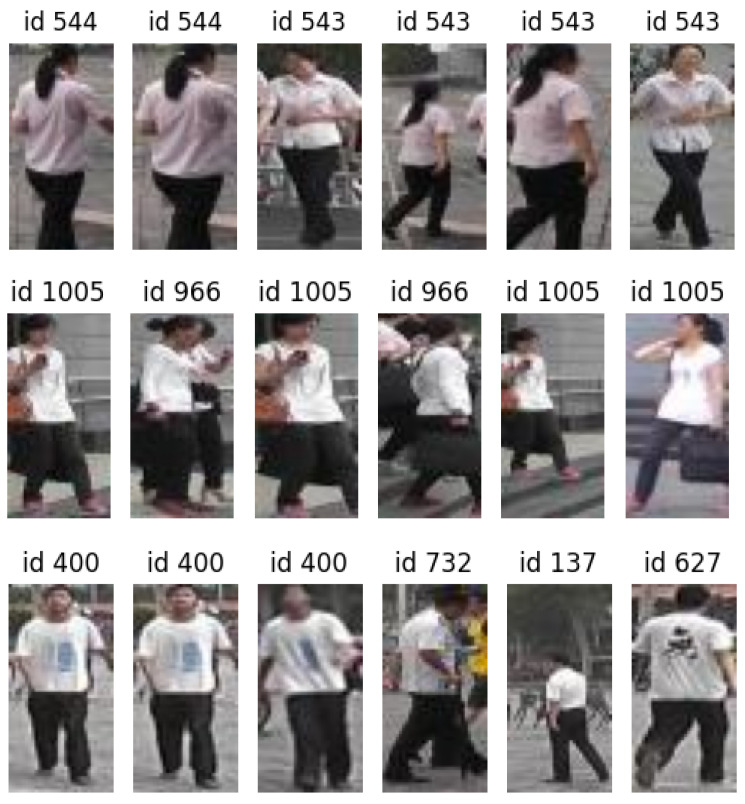
Person re-identification. In each row, the first image is the query, and the successive images are the 5 best matchings in the gallery.

**Figure 9 sensors-24-01240-f009:**
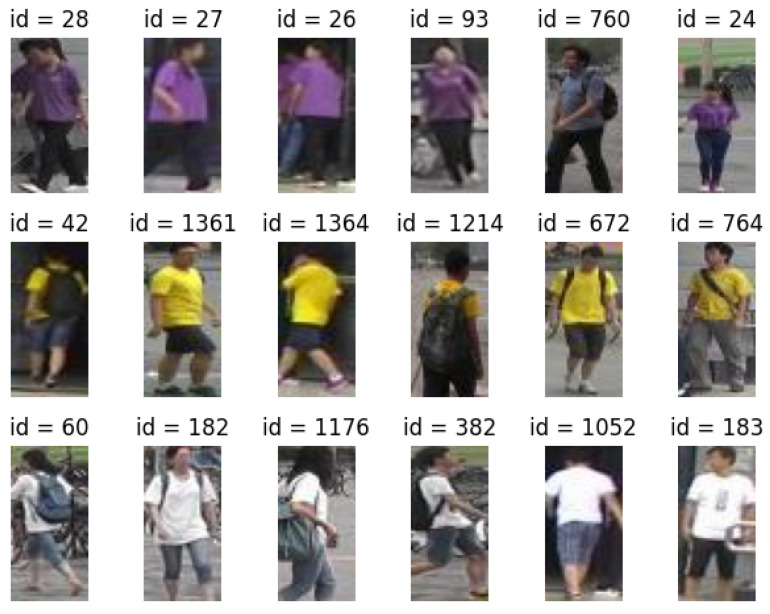
Inherent clustering of the latent space. For every ID in the initial column, we display five distinct individuals whose latent representations are closer, in terms of Euclidean distance, to the representation of the specified ID.

**Figure 10 sensors-24-01240-f010:**
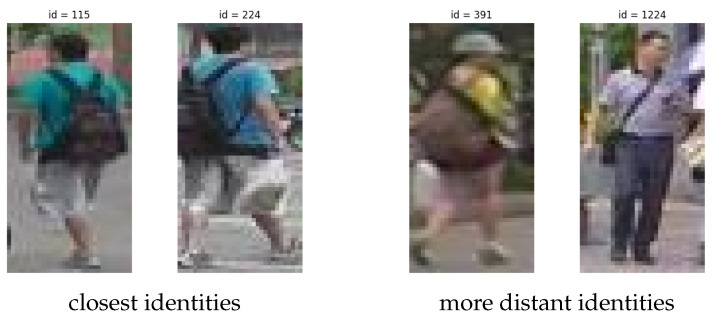
On the left, we have the two closest identities in the latent space of Market1501; on the right, the two more distant ones.

**Figure 11 sensors-24-01240-f011:**
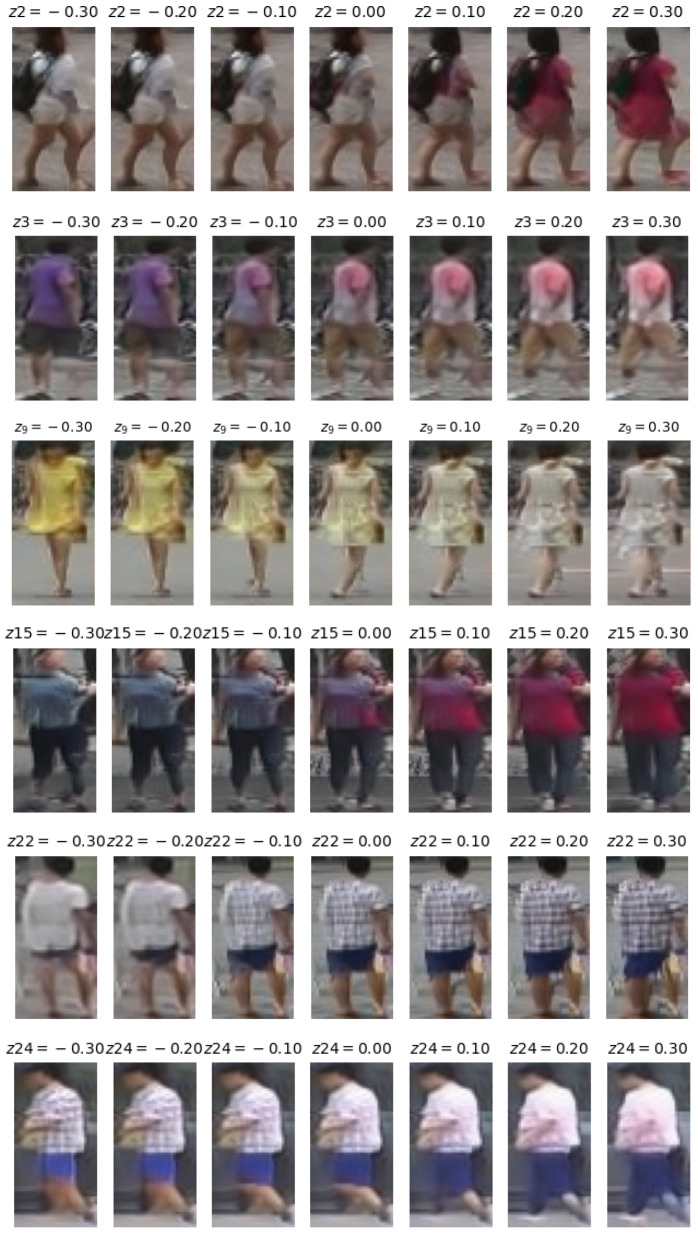
Effect of latent variables on generation. In each row, we keep the same random noise and vary a given variable in a predefined range. The specific variable and its value are indicated above the generated image.

## Data Availability

The application described in this paper is open-source. The software can be downloaded from the following github repository: https://github.com/asperti/GenerativePersonReID (accessed on 1 February 2024).

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
