# Peer review of "A Generative Approach to Person Reidentification"

_sensors, 2024, doi:10.3390/s24041240_

Round 1

Reviewer 1 Report

Comments and Suggestions for Authors

The submitted manuscript deals with person re-identification using diffusion models. The topic of the manuscript is a hot research topic in the computer vision literature and fits well to the scope of J. Imaging. In general, the manuscript uses a good English and has a logical structure.

I have the following concerns and comments related to the submitted manuscript.

i) The introduction section should be improved by pointwise listing the specific contributions of this study. Current statements are too general. The introduction section should have two subsections, i.e. Contributions and Structure of the paper. Now, the introduction section can be surveyed with difficulties due to the lack of articulation.

ii) Although vision based person re-identification is a hot research topic, the related work section is very short. The authors do not outline the main approaches. There is no organizing logic in this section. I think the authors should organize the reviewed methods based on the working principle. The authors mention contrastive learning for person re-identification (which is widely used in the literature), but no relevant methods were reviewed. For instance, authors could mention contrastive learning with multi-instance learning (Person Re-identification based on Deep Multi-instance Learning, 2017), with generative learning (Joint generative and contrastive learning for unsupervised person re-identification, 2021), and with attention mechanism (Mask-guided contrastive attention model for person re-identification, 2018). More recent methods can be also found: https://dblp.org/search?q=person%20re-identification

iii) The description of the proposed method is unfortunately rather weak. To tell the truth, I was not able to interpret Figure 4. Please draw first a general workflow of the proposed method. Subsequently, you can write about the details. But first the reviewer and readers should see the big picture.

iv) Unfortunately, the evaluation section is not in a good shape. First, you should describe the applied evaluation metrics and the benchmark database. Second, a comparison to relevant state-of-the-art cannot be avoided in such a hot research topic.

I think the manuscript is not suitable for publication. The authors should improve the manuscript at all major points, i.e. declaring contributions, related work section, description of the proposed method, experimental results. My recommendation is an extensive major revision.

Reviewer 2 Report

Comments and Suggestions for Authors

This paper propose a new method for person re-identification task.  By learning the representation of an individual as the conditioning information required to generate images of the specific person starting from random noise, it can extract the identity information of the individual. The idea sounds interesting. However, the paper doesn’t provide convincing results to show the superiority over STAR work. Here are some comments:

1.     We encourage the authors to express your idea as mathematical formulas.

2.     Algorithm1 and Algorithm2 are standard diffusion model. Does your method have any difference?

3.     The mean average precision (MAP) of the method is 73%, still far from the STAR 96%, the method still needs improvement.

4.     Figure 4 doesn’t have Caption.

5.     Figure 5 is too small. If you just use the U-net architecture, there is no need to put it here. We want to see your network.

Comments on the Quality of English Language

The Quality of English Language of the paper is OK.

Round 2

Reviewer 1 Report

Comments and Suggestions for Authors

I think the manuscript is in much better shape. The authors added new figures and explanations to the manuscript. Further, the structure of the paper was improved. I think the manuscript can be accepted now.